# Influence of Spraying Some Biostimulants on Yield, Fruit Quality, Oil Fruit Content and Nutritional Status of Olive (*Olea europaea* L.) under Salinity

Adel M. Al-Saif [1,*], Muhammad Moaaz Ali [2], Ahmed B. S. Ben Hifaa [3] and Walid F. A. Mosa [4]

[1] Department of Plant Production, College of Food and Agriculture Sciences, King Saud University, P.O. Box 2460, Riyadh 11451, Saudi Arabia

[2] College of Horticulture, Fujian Agriculture and Forestry University, Fuzhou 350002, China; muhammadmoaazali@yahoo.com

[3] Horticulture Department, Nasser's Faculty of Agricultural Sciences, University of Lahej, Lahej 73560, Yemen; ahmedbatam56@gmail.com

[4] Plant Production Department (Horticulture-Pomology), Faculty of Agriculture, Saba Basha, Alexandria University, Alexandria 21531, Egypt; walidmosa@alexu.edu.eg

* Correspondence: adelsaif@ksu.edu.sa

**Abstract:** Salinity currently affects more than 20% of agricultural land and is expected to pose potential challenges to land degradation and agricultural production in the future. It is a leading global abiotic stress that affects general plants and cultivated crops adversely. The utilization of biostimulants can enhance the efficiency of plant nutrition, facilitate the uptake of nutrients, boost crop yield, improve the quality characteristics of fruits and enhance plants' ability to withstand abiotic stresses. Biostimulants serve as a vital reservoir of macro- and microelements and plant hormones, such as auxins, cytokinins and gibberellins. Therefore, the current study was conducted to examine the effect of the foliar application of some biostimulants on relieving the side effects of salinity on olive trees (*Olea europaea*) cv. Kalamata. The olive trees were sprayed three times with moringa leaf aqueous extract (MLE) at 2, 4 and 6%, seaweed extract (SWE) at 1000, 2000 and 3000 ppm and their combinations: 2% MLE + 1000 ppm SWE (combination 1), 4% MLE + 2000 ppm SWE (combination 2) and 6% MLE + 3000 ppm SWE (combination 3). The results revealed that the application of biostimulants had a beneficial effect on the overall growth and development of olive trees, surpassing the performance of untreated trees. Spraying MLE and SWE, particularly at concentrations of 6% and 3000 ppm, respectively, significantly enhanced various aspects of olive tree performance. Notably, there were significant increases in leaf chlorophyll content, flower number, fruit set percentages, fruit yields, fruit oil content, fruit firmness, total soluble solid (TSS) percentage and leaf macro- and micronutrients. Furthermore, the combined application of MLE and SWE resulted in a greater effect when compared to using each one individually. In both seasons, combination 3 outperformed the other treatments that were applied.

**Keywords:** biostimulants; fruit quality; moringa; oil content; *Olea europaea*; seaweed

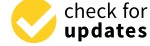



## 1. Introduction

In Egypt, the cultivated area of olive trees, olive (*Olea europaea* L.), spans 99,102 ha and produces 976,063 t. Globally, the cultivated area for olives is 103,382 ha, with a total production of 230,543 t [1]. Olive trees are the predominant fruit species in the Mediterranean basin, both in terms of the wide range of cultivated varieties and their social, economic and environmental significance [2–6]. According to Ribeiro-Gomes and Sacks [7], there are more than 805 million olive trees worldwide, with 98% of them concentrated in the Mediterranean region. The global olive genetic diversity is extensive, consisting of over 2600 distinct varieties [8].

Olive trees have adaptive morphological characteristics and physiological mechanisms to grow well and produce fruit under water scarcity, salinity and rainfall conditions in many arid and semiarid areas [9–11]. The olive is characterized by its low content of carbohydrates, fiber and protein, while its fruits have almost all essential amino acids. Moreover, olive oil contains many phenolic compounds, such as oleuropein, hydroxytyrosol, tyrosol, cinnamic acid, *p*-Coumaric acid and homovanillic acid [12], so olive fruits have many advantageous influences like antioxidant, antimicrobial and anti-inflammatory activities [13].

Salinity is a worldwide problem that causes a threat to sustainable agriculture and crop productivity. It has adverse effects on crop plants and could disrupt future food production [14–16]. When plants experience high levels of salinity stress, their ability to access and absorb essential nutrients, such as nitrogen, calcium, potassium, magnesium, phosphorus, nitrate, zinc and manganese, is substantially reduced, and this decrease is primarily caused by the combined influence of chloride and sodium ions, which interact and exacerbate the negative effects on nutrient availability and uptake and cause nutritional imbalances in both the soil and plant tissues [17–20]. Consequently, this imbalance results in metabolic disorders and compromised enzyme functionality, leading to adverse consequences on crucial biological processes, such as stomatal function, photosynthesis, germination, respiration, transpiration, plant growth and yield [21–26], phytohormone regulation and protein biosynthesis [27]. Therefore, there is a growing need to develop methods that can minimize the impact of salinity on crops.

Moringa leaf aqueous extract (MLE) is rich in zeatin, a natural cytokinin, as well as amino acids, proteins, phenolics, ascorbates, vitamin E and essential nutrients; hence, it is considered a plant growth regulator and an effective natural growth stimulator [28,29]. Furthermore, MLE aids in relieving the toxic impact of salt stress by improving plant growth characteristics [28]. MLE promotes flowering, enhances the quality of fruits and increases their yield [30]; its application helps mitigate the undesirable effects of salinity by enhancing proline, total sugar and phenol levels as well as catalase (CAT), ascorbate peroxidase (APX) and superoxide dismutase (SOD) activities [31,32]. The application of MLE has the ability to alleviate the side effects of environmental stresses and to raise the immunity of plants by enhancing plant physiological and biochemical characteristics [33]. Moreover, MLE has the advantage of being easily obtainable, cost-effective and eco-friendly, making it a promising candidate for utilization as a crop biostimulant [34]. MLE is rich in nutrients, including calcium, magnesium, potassium, manganese, zinc and iron, as well as proteins and amino acids [35]. Additionally, it was reported by Sun [36] that the application of MLE on grapefruit (*Citrus paradise* Macf.) resulted in the improvement of phenol, flavonoid and anthocyanin content. Zulfiqar [37] noticed that MLE acts as a growth and photosynthesis enhancer during salt stress [38]; therefore, its application has a contrary impact on modifying stress signaling, thus leading to raising the plant's tolerance to stress by balancing phytohormone levels. As a result, it promotes the plant's growth and its development. Abd El–Hamied and El-Amary [39] found that the use of MLE in pear cultivation resulted in enhanced yields as well as improvements in fruit size and weight. The treatment of grapes (*Vitis vinifera* L.) cv. Flame seedless with 2%, 4% and 6% MLE resulted in notable improvements in vine length, vine thickness, leaf chlorophyll levels, productivity and fruit quality when compared to the control treatment [40].

Seaweed extract (SWE) is an inexpensive and beneficial source of nutrients and plant growth stimulators, and its foliar spray has proven to be an effective approach for enhancing vegetative growth, photosynthetic rate, proline content and total soluble sugars in fruit crops, thus improving biotic and abiotic stress tolerance, increasing yield and fruit quality and extending the fruit's shelf life [41–45]. Moreover, it was mentioned that SWE is characterized by its high content of cytokinins, auxins [46] and polysaccharides, so it can improve plant growth [47]. The use of SWE has been found to positively impact leaf coloration by promoting chlorophyll biosynthesis or reducing chlorophyll degradation [48]. Additionally, these extracts stimulate the accumulation of photosynthetic pigments and

enhance leaf mineral content, particularly phosphorus and potassium [49]. Spraying SWE at concentrations of 0.2 and 0.4% on peach (*Prunus persica* (L.) Batsch) trees resulted in remarkable improvements in leaf area and elevated levels of chlorophyll content when compared to the control [50]. The foliar application of SWE improves the yield and quality of pears (*Pyrus communis* L.) [51] and strawberries (*Fragaria X ananassa* Duch.) [52]. Fornes et al. [53] stated that the exogenous foliar applications of SWE improved fruit yield by reducing the pre-mature fruit drop percentages in mandarin oranges (*Citrus reticulata* Blanco). Applying SWE to pear (*P. communis*) trees [54] and mangos (*Mangifera indica* L.) [55] resulted in higher leaf area, leaf chlorophyll content, photosynthetic rates, flower percentage and increased fruit yield. Spraying SWE on apples (*Malus domestica* Borkh) improved the development of shoots, leaves, the formation of flowers, fruit set percentage, fruit yield, fruit weight and size [56,57].

Consequently, this study is the first of its kind by investigating the effects of SWE and MLE, individually and in combination, as environmentally friendly biostimulants and exploring how these biostimulants can enhance the olive tree's resistance to salinity stress and improve its overall performance without any undesirable effects.

## 2. Materials and Methods

### 2.1. Characterization of the Experimental Area and Experimental Design

The present study was performed on 8-year-old olives cv. Kalamata, which were cultivated in a private orchard at Wady El Natron, Beheira Governorate, Egypt, during the 2021–2022 seasons. The olive trees were cultivated in sandy soil with a planting distance of $6 \times 6$ m, and irrigation was implemented using a drip irrigation system. The trees were fertilized with 48 $m^3$ manure, 240 kg of calcium superphosphate, 240 kg sulfur, 4 L of liquid phosphoric acid, 900 kg of ammonium nitrate, 1500 kg of ammonium sulfate, 600 kg potassium sulfate and 120 kg of magnesium sulfate per ha. Table 1 provides a description of the soil characteristics of the sandy soil in which the olive trees were cultivated [58]. The water quantity and chemical composition of the water utilized in the study is shown in Table 2. The experiment consisted of eighty trees with similar vigor, growth and size, and a randomized complete block design (RCBD) was employed for arranging the trees in the study, where each treatment consisted of eight replicates (trees). The trees within the experimental orchard were subjected to the same treatments.

**Table 1.** The composition of the experimental soil.

| Mechanical analysis | | | |
|---|---|---|---|
| Clay | Silt | Sand | Soil texture |
| 12.2% | 22.8% | 65% | Sandy loam |
| $CaCO_3^-$ | Organic matter | EC dSm$^{-1}$ (1:1) | pH (1:1) |
| 13% | 0.2% | 4.54 (Saline) | 8.3 |
| **Soluble cations and anions (meq/L)** | | | | | | |
| $Ca^{2+}$ | $Mg^{2+}$ | $Na^+$ | $K^+$ | $HCO_3^-$ | $Cl^-$ | $SO_4^{2-}$ |
| 14.40 | 6.80 | 13.05 | 9.38 | 16.06 | 13.78 | 12.65 |
| **Available macronutrients (mg/kg soil)** | | | **Available micronutrients (mg/L)** | | |
| N | P | K | Fe | Zn | Mn |
| 98 | 6.82 | 508 | 0.85 | 0.11 | 0.27 |

Olive trees were sprayed during the 2021–2022 seasons three times: middle of March, during full bloom (start of May) and the third spray was after three weeks of using MLE at 2, 4 and 6%, SWE at 1000, 2000 and 3000 ppm and their combinations, i.e., 2% MLE + 1000 ppm SWE (combination 1), 4% MLE + 2000 ppm SWE (combination 2) and 6% MLE + 3000 ppm SWE (combination 3), as compared to the untreated trees which were a

control treatment. At the beginning of the experiment, we performed a chemical analysis of both MLE and SWE. The composition of SWE was 48% organic matter; 3.0% N; 3.0% P; 18% $K_2O$; 0.2% Fe; 1.5% Zn; 0.5% B; 0.1 Mo; 18% alginic acid; 320–400 (mg/g) plant hormones. The composition of MLE was (mg/100 g); 208 P; 2100 K; 2225 Ca; 400 Mg; 900 S; 28 Fe; 0.5 Cu; 2.95 Se; 3700 carbohydrates; 2800 protein, 30 VB; 18.30 VC; 118.43 phenolic compounds. The effects of these treatments were evaluated on the following parameters:

**Table 2.** Water quantity and chemical composition of the water used in this study.

| Water quantity per tree (L/day) | | | | |
|---|---|---|---|---|
| January-February | March | April–September | October | November–December |
| 50 | 80 | 100 | 80 | 50 |
| **Water chemical composition of the used water** | | | | |
| Parameter | Sample | | | |
| **Textural class** | | | Micronutrients | |
| pH | 7.88 | | Fe | 0.39 mg/L |
| EC | 5.22 ds/m | | Zn | 0.02 mg/L |
| Salinity | 2067 ppm | | Mn | 0.03 mg/L |
| **Soluble cations** | | | Cu | 0.14 mg/L |
| $Na^+$ | 42.1 Meq/L | | **Soluble anions** | |
| $K^+$ | 0.55 Meq/L | | $Cl^-$ | 44.0 Meq/L |
| $Ca^+$ | 4.6 Meq/L | | $HCO_3^-$ | 5.20 Meq/L |
| $Mg^+$ | 3.8 Meq/L | | $CO_3^{2-}$ | - |
| | | | $SO_4^{2-}$ | 1.15 Meq/L |

### 2.2. Total Chlorophyll (SPAD)

The total chlorophyll content in fresh leaves was measured in SPAD units using a Minolta chlorophyll meter (SPAD-501) [59].

### 2.3. Flower Number, Fruit Set Percentages, Fruit Drop Percentages and Fruit Yield

Flower number: it was quantified per $m^2$. To determine the fruit set percentage and fruit drop as well as fruit yield, four branches were carefully selected from each side of every replicate (tree) and labeled. The number of flowers on each branch was counted, and as described by El-Hady et al. [60], the fruit set was calculated using the following Equation (1):

$$\text{Fruit set \%} = \frac{\text{number of fruit setting}}{\text{total number of flowers}} \times 100. \tag{1}$$

Fruit drop percentages were calculated using the following Equation (2):

$$\text{Fruit drop \%} = \frac{\text{number of fruit setting} - \text{number of mature fruits}}{\text{number of fruit setting}} \times 100. \tag{2}$$

Fruit yields: They were estimated for each replicate/tree in kg and in t per ha by multiplying the average of tree yield with the number of trees in one ha.

### 2.4. Fruit Quality

At the time of harvesting (October 2021–2022), 50 fruits from each replicate were chosen randomly and their weight, size, pulp weight and seed weight were measured by taking their average. Fruit length and diameter were estimated using a digital Vernier caliper (Cangxian Sanxing Hose Clamp Co., Ltd., Custom manufacturer, Cangzhou, China).

Fruit firmness was estimated using a Magness and Taylor pressure tester with a 7/18-inch plunger [61].

The moisture content of fruits was determined by measuring the initial weight of 50 fresh fruits; these fruits were then dried until a constant weight was reached and the moisture content was calculated as the difference between the initial weight and the final constant weight of the fruits. Total soluble solids were determined using a hand refractometer and the result was expressed as a percentage (%). Oil content: Samples from the flesh fruit were dried and then grinded, and 2 g were weighed, then filtered and placed in the Soxhlet apparatus using petroleum ether [62]. The oil percentage was calculated using the following Equation (3):

$$\text{Oil \%} = \frac{\text{weight of extracted oil}}{\text{weight of sample}} \times 100. \tag{3}$$

### 2.5. Leaf Chemical Composition

Following the harvesting period in November 2021–2022, thirty leaves were randomly selected from the middle part of the shoots in each replicate, as described by Arrobas et al. [63]. These selected leaves were then analyzed to determine their mineral content. The leaf samples were subjected to a series of preparation steps: First, they were washed with tap water and then rinsed with distilled water. Afterwards, the leaves were dried at 70 °C until a constant weight was achieved. Subsequently, the dried leaves were ground and subjected to acid digestion using $H_2SO_4$ and $H_2O_2$ until the solution became clear. Then, the digested solution was utilized for the analysis of nitrogen, phosphorus and potassium content. The nitrogen content (N) was determined using the micro Kjeldahl method [64]. Phosphorus content (P) was measured using the Vanadate-molybdate method [65]. Potassium content (K), on the other hand, was determined using a flame photometer following the method described by Asch et al. [66]. Leaf calcium (Ca), magnesium (Mg), iron (Fe), zinc (Zn) and manganese (Mn) concentrations were determined using atomic absorption spectroscopy, following the method described by Stafilov and Karadjova [67].

### 2.6. Statistical Analysis

The collected data were subjected to analysis of variance (ANOVA) using RCBD. Duncan's test was performed at a significance level of 0.05 to assess the differences among treatment means. The means were further compared using the least significant difference method at a probability of 5% [68]. The statistical analysis was conducted using CoHort Software version 6.311 (Pacific Grove, CA, USA).

## 3. Results

### 3.1. Total Chlorophyll, Flower Number and Fruit Set Percentages

The results presented in Table 3 demonstrated a significant enhancement in leaf total chlorophyll content, flower number and fruit set percentages in both the 2021 and 2022 seasons through the application of foliar sprays containing MLE, SWE and their combinations as compared to the control. These improvements were observed to be substantially higher when compared to untreated trees. The results demonstrated that the effectiveness of MLE and SWE increased with higher applied doses. Specifically, concentrations of 6% for MLE and 3000 ppm for SWE were found to be the most optimal during both experimental seasons. Among the different combinations tested, combination 3 exhibited the most favorable results, followed by combination 2.

**Table 3.** Effect of foliar spraying of MLE, SWE and their combinations on leaf total chlorophyll, flower number and fruit set percentages of olive cv. Kalamata during 2021–2022.

| Treatments | | Total Chlorophyll (SPAD) | | Flower Number (cm$^2$) | | Fruit Set % | |
|---|---|---|---|---|---|---|---|
| | | **2021** | **2022** | **2021** | **2022** | **2021** | **2022** |
| Control | 0 | 54.63 f | 57.18 f | 576.00 f | 671.00 g | 4.21 d | 4.39 f |
| MLE | 2% | 62.00 e | 62.00 e | 673.00 e | 711.00 f | 4.43 d | 4.70 ef |
| | 4% | 63.78 de | 67.47 d | 712.00 d | 757 e | 5.64 c | 5.73 bc |
| | 6% | 69.21 bc | 70.76 c | 752.00 c | 787 d | 5.85 bc | 6.04 ab |
| SWE | 1000 ppm | 63.26 de | 64.36 de | 686.00 e | 719.00 f | 4.34 d | 5.01 de |
| | 2000 ppm | 69.15 bc | 71.70 c | 752.00 c | 753.00 e | 566 c | 5.63 bc |
| | 3000 ppm | 72.87 b | 74.80 b | 797.00 b | 836 c | 6.12 ab | 6.14 ab |
| Combinations | 1 | 66.27 cd | 66.25 d | 673.00 e | 746 e | 5.55 c | 5.42 cd |
| | 2 | 72.87 b | 75.62 b | 798.00 b | 860.00 b | 6.51 a | 6.37 a |
| | 3 | 81.32 a | 81.96 a | 892.60 a | 936.00 a | 6.54 a | 6.49 a |
| LSD$_{0.05}$ | | 3.50 | 3.04 | 16.55 | 23.16 | 0.43 | 0.48 |

Duncan's test at 0.05 indicates that treatments labeled with the same letters did not have significant differences among them within each column.

### 3.2. Fruit Drop Percentage and Fruit Yield

Data in Table 4 indicated that the application of MLE and SWE statistically decreased the fruit drop percentages in the two seasons when compared with the control treatment. It was clear that the least percentage of the dopped fruits was accompanied by the spraying of combinations 3 and 2 as well as the spraying of 6% MLE or 3000 ppm SWE. On the other hand, the foliar spraying of MLE and SWE was effective in increasing the fruit yields as compared with unsprayed trees. The application of combination 3 was the most effective treatment that had the strongest positive effect on increasing the obtained yield in contrast with the other applied treatments and control. Additionally, combination 2 and 3000 ppm SWE significantly also increased the fruit yields over the application of combination 1, 2% MLE or 1000 ppm in the two seasons.

**Table 4.** Effect of foliar spraying of MLE, SWE and their combinations on fruit drop percentage; fruit production in kg or in a ton of olive cv. Kalamata during 2021–2022.

| Treatments | | Fruit Drop (%) | | Production (kg/Tree) | | Yield (t/ha) | |
|---|---|---|---|---|---|---|---|
| | | **2021** | **2022** | **2021** | **2022** | **2021** | **2022** |
| Control | 0 | 92.37 a | 89.47 a | 37.60 e | 39.96 g | 4.17 e | 4.43 g |
| MLE | 2% | 90.28 b | 86.59 b | 37.84 e | 41.14 fg | 4.20 e | 4.57 fg |
| | 4% | 87.73 cd | 83.51 cd | 41.96 d | 44.14 e | 4.66 d | 4.90 e |
| | 6% | 86.87 d | 82.46 d | 43.64 cd | 49.88 c | 4.84 cd | 5.54 c |
| SWE | 1000 ppm | 89.55 bc | 86.06 b | 37.90 e | 41.98 f | 4.20 e | 4.66 f |
| | 2000 ppm | 88.07 cd | 83.88 cd | 43.42 cd | 46.32 d | 4.82 cd | 5.14 d |
| | 3000 ppm | 86.56 d | 82.44 d | 44.86 bc | 51.28 bc | 4.98 bc | 5.69 bc |
| Combinations | 1 | 90.09 b | 84.22 c | 41.70 d | 44.58 e | 4.63 d | 4.95 e |
| | 2 | 84.08 e | 80.48 e | 46.46 b | 51.58 b | 5.16 b | 5.72 b |
| | 3 | 79.96 f | 78.17 f | 51.44 a | 53.96 a | 5.71 a | 5.99 a |
| LSD$_{0.05}$ | | 1.77 | 1.49 | 2.45 | 1.49 | 0.27 | 0.16 |

Duncan's test at 0.05 indicates that treatments labeled with the same letters did not have significant differences among them within each column.

### 3.3. Fruit Quality

According to the results presented in Table 5, the application of MLE and SWE via spraying led to notable enhancements in the physical characteristics of olive cv. Kalamata

during the 2021–2022 period. Specifically, fruit weight, size, length and diameter exhibited significant improvements when treated with 4 or 6% MLE, 2000 or 3000 ppm SWE as well as combinations of 2 or 3 of these biostimulants. Significant increases in fruit weight, size, length and diameter were observed by applying combination 3, followed by combination 2, during the 2021–2022 seasons. Furthermore, it was demonstrated that the effects of MLE and SWE increased gradually as the applied doses were raised, where 6% MLE showed superior results when compared to 4 or 2% concentrations, and 3000 ppm SWE outperformed 2000 or 1000 ppm concentrations.

**Table 5.** Effect of foliar spraying of MLE, SWE and their combinations on fruit weight, size, length and diameter of olive cv. Kalamata during 2021–2022.

| Treatments | | Fruit Weight (g) | | Fruit Size (cm$^3$) | | Fruit Length (cm) | | Fruit Diameter (cm) | |
| --- | --- | --- | --- | --- | --- | --- | --- | --- | --- |
| | | **2021** | **2022** | **2021** | **2022** | **2021** | **2022** | **2021** | **2022** |
| Control | 0 | 5.28 f | 5.76 f | 6.41 f | 6.94 f | 2.40 e | 2.70 g | 1.70 e | 1.67 f |
| MLE | 2% | 5.49 ef | 5.83 f | 6.67 f | 7.03 f | 2.86 c | 2.90 f | 1.72 e | 1.81 e |
| | 4% | 6.41 d | 6.60 cde | 7.63 cd | 7.54 d | 2.91 c | 3.05 e | 1.86 d | 1.99 cd |
| | 6% | 6.88 bcd | 6.76 cd | 8.05 bc | 7.85 cd | 3.22 b | 3.31 d | 1.97 c | 2.07 bc |
| SWE | 1000 ppm | 5.64 ef | 5.98 f | 6.78 ef | 7.47 de | 2.68 d | 2.84 f | 1.85 d | 1.89 de |
| | 2000 ppm | 6.52 cd | 6.51 de | 7.77 bcd | 7.71 d | 3.27 b | 3.34 d | 1.89 d | 1.92 de |
| | 3000 ppm | 6.96 bc | 6.90 c | 8.19 bc | 8.18 c | 3.31 b | 3.47 c | 2.12 b | 2.08 bc |
| Combinations | 1 | 5.91 e | 6.31 e | 7.24 de | 7.13 ef | 3.01 c | 306 e | 1.85 d | 1.86 e |
| | 2 | 7.09 b | 7.47 b | 8.23 b | 8.97 b | 3.39 b | 3.61 b | 2.05 bc | 2.18 b |
| | 3 | 7.81 a | 8.35 a | 9.13 a | 9.61 a | 3.60 a | 3.76 a | 2.23 a | 2.39 a |
| LSD$_{0.05}$ | | 0.50 | 0.32 | 0.52 | 0.39 | 0.16 | 0.11 | 0.08 | 0.10 |

Duncan's test at 0.05 indicates that treatments labeled with the same letters did not have significant differences among them within each column.

Results in Table 6 showed that pulp fruit, seed weights and fruit firmness were increased by the application of MLE at 6 or 4%, 2000 or 3000 ppm and their different combinations in the two seasons as compared to the control. Furthermore, the highest increases were observed when applying combination 3, followed by combination 2, in both seasons. In terms of the flesh-to-fruit ratio, it was noticed that the most pronounced increase was accompanied by spraying combination 3 and then combination 2, 3000 ppm SWE and 6% MLE in both 2021–2022 seasons as compared with the control. Conversely, the results revealed that the application of combinations 3 and 2 significantly reduced fruit moisture content when compared to combination 1 in both experimental seasons. Furthermore, the application of 3000 or 2000 ppm SWE, along with 6 or 4% MLE, proved to be more effective in reducing fruit moisture content when compared to 1000 ppm SWE or 2% MLE in the 2021–2022 seasons.

The results displayed in Table 7 provide clear evidence that the application of combinations 3 and 2, along with the use of 3000 ppm SWE and 6% MLE, had a substantial positive impact on fruit quality attributes, including total soluble solid percentages (TSS %) and fruit oil content, over unsprayed trees in both experimental seasons. Furthermore, in experimental seasons, the application of 2000 ppm SWE and 4% MLE enhanced TSS percentage and fruit oil content when compared to untreated trees.

**Table 6.** Effect of foliar spraying of MLE, SWE and their combinations on fruit pulp weight, seed weight, pulp–fruit ratio, fruit firmness and moisture content of olive cv. Kalamata during 2021–2022.

| Treatments | | Pulp Weight (g) | | Seed Weight (g) | | Pulp–Fruit Ratio | | Fruit Firmness (Ib/inch$^2$) | | Moisture Content% | |
|---|---|---|---|---|---|---|---|---|---|---|---|
| | | 2021 | 2022 | 2021 | 2022 | 2021 | 2022 | 2021 | 2022 | 2021 | 2022 |
| Control | 0 | 4.04 f | 4.53 g | 1.23 ab | 1.23 de | 0.77 e | 0.78 e | 4.44 e | 4.55 e | 68.8 a | 72.10 a |
| MLE | 2% | 4.37 ef | 4.63 g | 1.13 b | 1.19 e | 0.79 bcd | 0.79 de | 4.56 e | 5.11 d | 65.39 b | 70.23 b |
| | 4% | 4.79 de | 5.32 de | 1.31 a | 1.29 bcd | 0.79 bcd | 0.80 cd | 5.32 c | 5.50 cd | 61.84 cd | 67.38 cd |
| | 6% | 5.47 bc | 5.48 cd | 1.35 a | 1.28 bcd | 0.80 bc | 0.81 bc | 5.74 b | 5.56 c | 61.87 cd | 64.42 e |
| SWE | 1000 ppm | 4.44 ef | 4.76 fg | 1.24 ab | 1.22 de | 0.78 de | 0.79 de | 4.90 d | 5.40 cd | 63.55 bc | 68.60 c |
| | 2000 ppm | 5.03 cd | 5.18 de | 1.35 a | 1.32 abc | 0.79 bcd | 0.80 de | 5.38 c | 5.46 cd | 61.47 cd | 67.05 d |
| | 3000 ppm | 5.51 bc | 5.65 c | 1.30 a | 1.25 cde | 0.82 ab | 0.82 b | 5.62 b | 6.23 b | 59.92 d | 63.90 e |
| Combinations | 1 | 4.66 de | 5.03 ef | 1.25 ab | 1.28 bcd | 0.79 cde | 0.80 de | 4.82 d | 5.36 cd | 62.58 c | 67.97 cd |
| | 2 | 5.61 b | 6.13 b | 1.36 a | 1.34 ab | 0.81 bc | 0.82 b | 5.68 b | 6.07 b | 56.67 e | 61.12 f |
| | 3 | 6.29 a | 6.99 a | 1.31 a | 1.37 a | 0.83 a | 0.84 a | 6.20 a | 6.70 a | 54.38 f | 58.26 g |
| LSD$_{0.05}$ | | 0.46 | 0.31 | 0.14 | 0.07 | 0.02 | 0.01 | 0.22 | 0.39 | 1.99 | 1.33 |

Duncan's test at 0.05 indicates that treatments labelled with the same letters did not have significant differences among them within each column.

**Table 7.** Effect of foliar spraying of MLE, SWE and their combinations on fruit content from TSS % and fruit oil content of olive cv. Kalamata during 2021–2022.

| Treatments | | TSS (%) | | Oil Content (%) | |
|---|---|---|---|---|---|
| | | 2021 | 2022 | 2021 | 2022 |
| Control | 0 | 12.45 f | 13.20 d | 13.19 g | 14.47 g |
| MLE | 2% | 13.12 def | 13.71 cd | 13.63 g | 15.06 g |
| | 4% | 13.32 cde | 13.98 c | 15.43 e | 17.63 d |
| | 6% | 13.83 cd | 14.72 b | 16.50 cd | 18.03 cd |
| SWE | 1000 ppm | 13.22 cde | 13.82 c | 14.57 f | 15.72 f |
| | 2000 ppm | 13.81 cd | 14.00 c | 15.94 de | 17.78 cd |
| | 3000 ppm | 14.64 b | 15.04 b | 17.01 bc | 18.32 bc |
| Combinations | 1 | 12.65 ef | 13.80 c | 15.37 e | 16.96 e |
| | 2 | 13.89 c | 14.88 b | 17.52 b | 18.81 b |
| | 3 | 15.37 a | 15.81 a | 18.50 a | 20.43 a |
| LSD$_{0.05}$ | | 0.67 | 0.53 | 0.60 | 0.62 |

Duncan's test at 0.05 indicates that treatments labelled with the same letters did not have significant differences among them within each column.

### 3.4. Nutritional Status

3.4.1. Leaf Mineral Content from Nitrogen, Phosphorous and Potassium

The results presented in Table 8 indicate that the application of combination 3 resulted in significant improvements in the leaf mineral content of macronutrients, such as nitrogen, phosphorous and potassium. This treatment exhibited the most pronounced increments in the mentioned nutrients when compared to the other treatments applied. Furthermore, the spraying of combination 2 showed significant effectiveness in increasing the fruit content of nitrogen, phosphorus and potassium when compared to the application of combination 1 in both seasons. Additionally, during both study seasons, the application of 2000 ppm SWE or 4% MLE demonstrated significant effectiveness in increasing the leaf content of nitrogen, phosphorus and potassium. These concentrations proved to be more effective when compared to the application of 1000 ppm SWE or 2% MLE in terms of nutrient uptake.

**Table 8.** Effect of foliar spraying of MLE, SWE and their combinations on leaf content from nitrogen, phosphorous and potassium of olive cv. Kalamata during 2021–2022.

| Treatments | | Nitrogen (%) | | Phosphorous (%) | | Potassium (%) | |
|---|---|---|---|---|---|---|---|
| | | **2021** | **2022** | **2021** | **2022** | **2021** | **2022** |
| Control | 0 | 1.41 f | 1.46 e | 0.39 f | 0.40 e | 0.95 f | 0.99 e |
| MLE | 2% | 1.43 f | 1.48 e | 0.43 e | 0.43 de | 1.01 e | 1.01 e |
| | 4% | 1.49 de | 1.54 d | 0.47 d | 0.49 c | 107 d | 1.10 d |
| | 6% | 1.51 d | 1.61 c | 0.49 cd | 0.52 b | 1.12 c | 1.13 cd |
| SWE | 1000 ppm | 1.47 e | 1.49 e | 0.46 d | 0.43 de | 1.00 e | 1.03 e |
| | 2000 ppm | 1.48 de | 1.55 d | 0.47 d | 0.49 c | 1.08 d | 1.12 cd |
| | 3000 ppm | 1.58 c | 1.67 b | 0.51 bc | 0.52 b | 1.17 b | 1.16 c |
| Combinations | 1 | 1.47 e | 1.48 e | 0.46 d | 0.44 d | 1.06 d | 1.09 d |
| | 2 | 1.63 b | 1.65 bc | 0.52 b | 0.54 b | 1.15 b | 1.22 b |
| | 3 | 1.76 a | 1.78 a | 0.54 a | 0.57 a | 1.24 a | 1.30 a |
| LSD$_{0.05}$ | | 0.03 | 0.04 | 0.02 | 0.03 | 0.04 | 0.04 |

Duncan's test at 0.05 indicates that treatments labeled with the same letters did not have significant differences among them within each column.

### 3.4.2. Leaf Mineral Content from Boron, Zinc, Iron and Manganese

The results concerning the effect of MLE and SWE on the leaf mineral content of macronutrients, such as boron, zinc, iron and manganese, are listed in Table 9. The foliar application of MLE, SWE and their various concentrations exhibited varying effects, with the most substantial increases observed when spraying combination 3, followed by combination 2, in both seasons. Furthermore, the results demonstrated that the application of 3000 ppm SWE and 6% MLE resulted in significant and noticeable increases in the leaf mineral content of boron, zinc, iron and manganese when compared to the application of 1000 ppm SWE or 2% MLE in both seasons.

**Table 9.** Effect of foliar spraying of MLE, SWE and their combinations on leaf content of boron, zinc, iron and manganese of olive cv. Kalamata during 2021–2022.

| Treatments | | Boron (ppm) | | Zinc (ppm) | | Iron (ppm) | | Manganese (ppm) | |
|---|---|---|---|---|---|---|---|---|---|
| | | **2021** | **2022** | **2021** | **2022** | **2021** | **2022** | **2021** | **2022** |
| Control | 0 | 41.51 e | 45.22 d | 30.79 f | 33.19 f | 102.23 f | 103.73 f | 31.52 g | 33.90 d |
| MLE | 2% | 44.33 e | 45.32 d | 31.11 f | 34.60 ef | 103.10 ef | 104.17 f | 33.74 f | 35.42 d |
| | 4% | 48.68 d | 51.52 c | 36.03 cd | 36.70 de | 107.23 cd | 107.43 cd | 36.22 e | 38.76 c |
| | 6% | 53.64 c | 52.18 c | 36.15 cd | 37.58 cd | 109.40 bc | 111.10 bc | 38.97 cd | 41.76 b |
| SWE | 1000 ppm | 47.68 d | 46.57 d | 32.83 ef | 34.53 ef | 105.63 de | 105.70 ef | 35.44 e | 35.38 d |
| | 2000 ppm | 51.68 c | 52.42 c | 34.53 de | 38.11 cd | 108.63 bcd | 108.77 cd | 37.92 d | 39.72 c |
| | 3000 ppm | 54.37 c | 53.87 c | 38.16 bc | 39.34 c | 110.13 bc | 112.50 b | 39.81 bc | 42.58 b |
| Combinations | 1 | 47.83 d | 47.31 d | 32.93 ef | 35.38 ef | 105.40 de | 106.33 def | 35.14 ef | 37.87 c |
| | 2 | 57.88 b | 60.17 b | 39.17 b | 42.19 b | 111.23 b | 113.03 b | 40.55 b | 42.51 b |
| | 3 | 63.10 a | 64.60 a | 42.73 a | 46.06 a | 115.13 a | 116.97 a | 42.29 a | 45.1 a |
| LSD$_{0.05}$ | | 3.00 | 3.14 | 2.45 | 2.07 | 3.00 | 2.53 | 1.42 | 1.98 |

Duncan's test at 0.05 indicates that treatments labeled with the same letters did not have significant differences among them within each column.

### 4. Discussion

According to the obtained results, the application of the highest concentrations and combinations of MLE and SWE had positive effects on leaf total chlorophyll, yield, oil content and fruit quality, as well as leaf mineral content of macro- and micronutrients in the two experimental seasons. Moreover, they increased the resistance of olive trees to salinity

because of their nutritional content, which led to improving the absorption of nutrients. These results were previously explained by many authors; they reported that in saline soil, increasing the salinity usually decreases the absorption of nutrients from the soil and, consequently, it negatively affects vegetative growth, yield, fruit oil content and the fruit quality of olive trees [16,20,69,70]. Moreover, Gopalakrishnan et al. [34] stated that moringa (*Moringa oleifera* Lam.) is known for its rich nutrient content, including vitamins, β-carotene, flavonoids, phenolic acids and fatty acids. Additionally, MLE acts as a natural biostimulant due to its composition of hormones, like auxins, gibberellins, cytokinins, sugars, tannins, proline, flavonoids, sterols, tannins, proteins, minerals, vitamins, essential amino acids, glucosinolates, isothiocyanates, phenolics and ascorbates [71–73]. Therefore, applying MLE resulted in enhancing plant growth and the levels of nutrients, such as nitrogen, phosphorus, potassium, iron, calcium and magnesium in the leaves, as well as improving fruit set percentage, fruit quality and quantity and yield characteristics [34,74]. Additionally, it was also reported that MLE plays a crucial role in improving plant nutrition, seed germination, vegetative growth, flowering level, photosynthetic rate, fruiting, gas exchange rates, water content regulation and utilization efficiency. Hence, it can increase root growth, yield components and fruit quality traits, particularly under stressful conditions like salinity, drought and heavy metals, by improving the activity of antioxidant enzymes and sugar content [38,75,76]. The application of MLE reduced the percentage of fruit drop in pears (*P. communis*) trees [39] and in mandarins (*C. reticulata*) [77]. In the same trend, Hassan et al. [78] stated that the application of MLE at concentrations of 2% and 4% on olive (*O. europaea*) trees increased fruit number, leaf area and leaf mineral contents of nitrogen, phosphorous, potassium, fruit set percentage, yield and fruit oil content when compared to a control. Mahmoud et al. [79] found that spraying plums (*Prunus domestica* L.) cv. Hollywood with 4%, 5% and 6% MLE at different growth stages improved fruit set, yield, weight, firmness, color, soluble solid content, ascorbic acid and anthocyanin content. It also reduced fruit drop and titratable acidity. Among the treatments, the 6% MLE spray was the most effective in enhancing these characteristics when compared to the other concentrations used. Spraying MLE at 2, 4 and 6% on peach (*P. persica*) cv. Early Grand trees increased yield, fruit diameter, fruit weight, pulp, pulp/stone ratio, TSS, TSS-acidity ratio and total, reducing and non-reducing sugars, as well as vitamin C levels, while it minimized the percentages of fruit drop and total acidity [80]. The application of 4 and 6% MLE on apple trees (*M. domestica*) cv. Anna exhibited notable increases in shoot length, shoot diameter, leaf chlorophyll content, fruit set, fruit yield, fruit weight, fruit size, soluble solids content, total sugar content and the leaf content of macronutrients when compared to untreated trees [81].

SWE is often considered a biostimulant due to its high content of cytokinin, auxin and gibberellins [46,82,83]. When SWEs are applied to plants, they can have several beneficial effects, such as increasing the total chlorophyll content of the leaves in treated plants; this improves the photosynthetic process, efficiency and plant growth [49,84,85]. SWEs are believed to boost the internal synthesis of polyamines and inhibit their breakdown, which likely contributes to the observed improvements in plant growth and productivity [48,86,87]. It was documented that applying SWE on date palms promoted their growth and characteristics, like total dry matter, leaf area, stomatal conductance and nitrogen and phosphorus content [88]. SWE is known for its rich content of organic material, micronutrients, macronutrients, vitamins, cytokinins and auxins. Consequently, these components play a crucial role in increasing crop yield by improving nutrient uptake and promoting the growth of roots, leaves, flowers and fruits. SWE also enhances soil structure, productivity and microbiological content, while stimulating the growth of beneficial soil microbes and the production of soil-conditioning substances. Additionally, SWEs can enhance plants' tolerance to abiotic stresses like salinity, cold or drought [89–93]. Furthermore, it has been stated by many authors that SWE is rich in macro- and micronutrients, such as Ca, C, Mg, P, K, S, B, Co, Fe, Mn, Mo, Se, Si and Zn [94,95]; therefore, its application can enhance plants' uptake of nutrients from the soil [96–98]. Spraying mangos (*M. indica*) with SWE

at 1, 2, 3 and 4% increased the number of fruits, fruit set percentage and fruit retention, number of fruits per panicle, number of fruits/tree, fruit yields in kilograms or tons and marketable fruit; 1% was the best treatment over the rest of the applied treatments [99]. The application of SWE on apples (*M. domestica*) cv. Gala at 0.1, 0.2, 0.3, 0.4 and 0.6% increased the percentage of fruit set percentage, fruit number, weight and length over control [100]. Spraying apple (*M. domestica*) cv. Anna with 0.3 or 0.4% SWE led to notable improvements in shoot length, shoot diameter, leaf chlorophyll content, fruit set, fruit yield, fruit weight, fruit size, soluble solids content, total sugar content, reduced sugar content and the leaves' content of macronutrients when compared to untreated trees [81]. Additionally, SWE also greatly improved fruit set percentage, fruit number, fruit retention, yield, fruit firmness and flesh and fruit skin color in avocados (*Persea americana* Mill.) cvs. Hass and Shepard [101]. It was reported by many authors that spraying SWE was more effective in enhancing vegetative growth, yield, fruit chemical and physical characteristics and the nutritional status of grapes when compared to untreated trees [102–104]. In previous studies, it was found that treating orange trees with SWE led to an increase in the maturity index and yield, while simultaneously reducing fruit drop [105,106]. Al-Saif et al. [107] found that spraying apricot trees with SWE at concentrations of 1000, 2000 and 3000 ppm resulted in noteworthy enhancements in various aspects like shoot length, leaf area, leaf chlorophyll content, fruit set, fruit yields and macro- and micronutrients in the leaves as well as the physical and chemical characteristics of the fruit. Notably, the application of a 3000 ppm concentration demonstrated superior results when compared to the other concentrations, showing greater improvements in these parameters.

Moisture content decreased in parallel to the increase in the oil content in the fruits. There is an inverse relationship between the oil content of fruits and their moisture content. This was previously confirmed by many authors, who stated that moisture content decreases while oil content increases until a certain level of maturity is reached [108–110].

## 5. Conclusions

The current study proved that the application of 6% MLE + 3000 ppm SWE increased the growth, yield, fruit quality traits and nutritional status of olive under salinity conditions when compared to the application of MLE or SWE individually; the combinations of 2% MLE + 1000 ppm SWE, 4% MLE + 2000 ppm SWE; and the control. Our results suggest that the combination of the biostimulants MLE and SWE could be utilized as an eco-friendly alternative to reduce full dependency on the use of chemical fertilizers in olive orchards under salinity to improve the production quality and maintain soil characteristics.

**Author Contributions:** Conceptualization, W.F.A.M. and A.B.S.B.H.; methodology, W.F.A.M.; software, W.F.A.M., A.M.A.-S. and A.B.S.B.H.; validation, A.M.A.-S., M.M.A. and A.B.S.B.H.; formal analysis, W.F.A.M., M.M.A., A.M.A.-S. and A.B.S.B.H.; investigation, W.F.A.M., resources, W.F.A.M., A.M.A.-S. and A.B.S.B.H.; data curation, W.F.A.M., A.M.A.-S. and A.B.S.B.H.; writing—original draft preparation, W.F.A.M., A.B.S.B.H. and M.M.A.; writing—review and editing, W.F.A.M., A.B.S.B.H., A.M.A.-S. and M.M.A.; supervision, W.F.A.M. and M.M.A. All authors have read and agreed to the published version of the manuscript.

**Funding:** This research was funded by Researchers Supporting Project number (RSP2023R334), King Saud University, Riyadh, Saudi Arabia.

**Data Availability Statement:** All the required data are inserted in the manuscript.

**Acknowledgments:** The authors extend their appreciation to the Researchers Supporting Project number (RSP2023R334), King Saud University, Riyadh, Saudi Arabia.

**Conflicts of Interest:** There are no conflicts of interest among all the authors.

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
