# Peer review of "Influence of Spraying Some Biostimulants on Yield, Fruit Quality, Oil Fruit Content and Nutritional Status of Olive (Olea europaea L.) under Salinity"

_horticulturae, doi:10.3390/horticulturae9070825_

Round 1

Reviewer 1 Report

Dear Authors,

I think that the study you have prepared will make useful contributions to the literature. I marked some suggestions and minor corrections on the text. Also, there is a lot of data in the Results and it would be helpful to discuss them more broadly in the discussion section with more recent references.

Kind regards

Reviewer 2 Report

The authors have submitted a manuscript in which they analyze the effectiveness of the application of some biostimulant products in the alleviation of the negative outcomes of salinity stress.

The topic of this paper is suitable for the scope of the Journal.

The Introduction is well presented. However, the novelty is not well explained. Are there other methods that can minimize the impact of salinity on crops? Please, cite them briefly. Is this the first study on the application of MLE and seaweed for the alleviation of the negative outcomes of salinity stress?

Paragraph 2.1: On how many plants was the experimental plan carried out? How were the plants to be sprayed with the respective biostimulant chosen?

Discussion must be improved, as it is too general. You mentioned the composition of biostimulants (lines 271, 291, 313), but can you have a more detailed discussion? For example, which of these compounds could have the greatest impact on each of the parameters analyzed in the results? Furthermore, results of other works on very different crops are cited. Are there specific data on the olive tree?

The data shows that greater applications of biostimulants improve the data analysed, but this suggests a higher cost. Are the results obtained in terms of yield high enough to guarantee a positive cost/benefit ratio?

The conclusions need to be improved. What are the current limits for a large-scale application of these biostimulants? Are there commercial products already in use? What further studies are needed? Can the composition and/or dosage of biostimulants be optimised?

Revise English grammar. Some examples:

e.g. Line 32-33 Not clear, revise English

Line 48 Capital letter after full stop.

Line 63 First time MLE is used. Explain the acronym

Line 97 2022

Reviewer 3 Report

The manuscript entitled "Influence of Spraying Some Biostimulants on Yield, Fruit Quality, Oil Fruit Content and Nutritional Status of Olive" presents important and interesting results for olive tree production under conditions of soil salinity. However, the manuscript needs substantial improvements in writing, structure and details for publication merit, as suggested in the attached file. In addition, the manuscript requires a detailed review of the English language by companies indicated by the MDPI. After accepting all the suggestions, I believe that the manuscript is ready and I request you to review the manuscript to verify the corrections.

The manuscript entitled "Influence of Spraying Some Biostimulants on Yield, Fruit Quality, Oil Fruit Content and Nutritional Status of Olive" presents important and interesting results for olive tree production under conditions of soil salinity. However, the manuscript needs substantial improvements in writing, structure and details for publication merit, as suggested in the attached file. In addition, the manuscript requires a detailed review of the English language by companies indicated by the MDPI. After accepting all the suggestions, I believe that the manuscript is ready and I request you to review the manuscript to verify the corrections.

Round 2

Reviewer 2 Report

I thank the authors for their kind answers

Reviewer 3 Report

The manuscript still needs minor modifications, but most of the suggestions have been carried out. Thanks

The manuscript still needs minor modifications, but most of the suggestions have been carried out. Thanks
